# Genome-Wide Identification, Characterization, and Expression Analysis Related to Low-Temperature Stress of the *CmGLP* Gene Family in *Cucumis melo* L.

**DOI:** 10.3390/ijms23158190

**Published:** 2022-07-25

**Authors:** Zhengda Zhang, Yongshuai Wen, Luqiao Yuan, Yuhui Zhang, Jingyi Liu, Fan Zhou, Qunning Wang, Xiaohui Hu

**Affiliations:** 1College of Horticulture, Northwest A&F University, Xianyang 712100, China; zhengda_zhang@163.com (Z.Z.); wen1102180281@163.com (Y.W.); wjswylq@163.com (L.Y.); zyh_0065@163.com (Y.Z.); ljy584233912@163.com (J.L.); zf3223471361@163.com (F.Z.); m18082957467@163.com (Q.W.); 2Key Laboratory of Protected Horticultural Engineering in Northwest, Ministry of Agriculture, Xianyang 712100, China; 3Shaanxi Protected Agriculture Research Centre, Xianyang 712100, China

**Keywords:** *Cucumis melo* L., *GLP* gene family, gene structure, low-temperature stress

## Abstract

Germin-like protein (GLP) participates in plant growth and development and plays an important role in plant stress. In the present study, 22 *CmGLPs* belonging to five classes were identified in the melon genome. Each member of the CmGLPs family contains a typical Cupin_1 domain. We conducted a genome-wide analysis of the melon *GLP* gene family characterization. *CmGLPs* were randomly distributed in the melon chromosomes, with the largest number on chromosome 8, having eight family members. Gene duplication events drive the evolution and expansion of the melon *GLP* gene family. Based on the phylogenetic tree analysis of GLP proteins in melon, rice, Arabidopsis, and cucumber, it was found that the *GLP* gene families of different species have diverged in evolution. Based on qRT-PCR results, all members of the *CmGLP* gene family could be expressed in different tissues of melon. Most *CmGLP* genes were up-regulated after low-temperature stress. The relative expression of *CmGLP2-5* increased by 157.13 times at 48 h after low-temperature treatment. This finding suggests that the *CmGLP2-5* might play an important role in low-temperature stress in melon. Furthermore, quantitative dual LUC assays indicated that *CmMYB23* and *CmWRKY33* can bind the promoter fragment of the *CmGLP2-5*. These results were helpful in understanding the functional succession and evolution of the melon *GLP* gene family and further revealed the response of *CmGLPs* to low-temperature stress in melon.

## 1. Introduction

Germin was originally described as a specific marker of wheat germination and was named “germin” because of its function [1]; since then, it has been found in many plants, including monocotyledons and dicotyledons, such as *Oryza sativa* L. [2] and *Arabidopsis* [3]. Germin-like proteins (GLP) are homohexameric glycoproteins that belong to the Cupin superfamily and share 30–70% sequence identity with germin protein [2,4]. GLP generally consists of two exons and one intron, encodes a protein consisting of approximately 220 amino acids, and contains a conserved Cupin domain with metal ion binding at the C-terminus [5,6,7]. The functions of GLPs are diverse. Most GLP proteins show enzyme activities in the form of polymers, and the enzyme functions are similar to superoxide dismutase (SOD), oxalate oxidase (OXO), and ADP glucose pyrophosphatase/phosphodiesterase (AGPPase) activities [2,8,9]. SOD can catalyze the conversion of superoxide into hydrogen peroxide and oxygen [10]. Plants can further convert hydrogen peroxide into other harmless substances [11]. OXO detoxifies by oxidizing oxalic acid to produce H_2_O_2_ and CO_2_, and plays an important role in plant disease resistance defense, growth, and development [12]. AGPPase is a major player in the transfer of carbon flow from starch synthase and may play a key role in starch biosynthesis [13]. Therefore, GLP proteins may perform the functions of these three enzymes. In addition to their enzymatic activities, *GLPs* are involved in plant life activities as structural proteins or receptors [14].

GLPs can be expressed in various tissues of plants and can play an important role in plant growth and development, such as plant growth, seed dormancy, and fruit development [4,15,16]. *PDGLP* regulates primary root growth by controlling phloem-mediated resource allocation between primary and lateral root meristems [3]. Rice activates the *OsGLP* by regulating the gibberellin signaling pathway, thereby regulating seed primary dormancy [4]. GLPs are also involved in plant responses to abiotic stresses [2,17]. The results of transcriptome verification show that potato improved tolerance to salt stress by upregulating the expression of *StGLP5* [7]. After knocking out the *OsGLP1*, rice became sensitive to UV-B stress and downregulated the expression of some UV-B protective genes, indicating that *OsGLP1* is involved in the adaptation of rice to UV-B radiation [2]. GLP also participates in the defense mechanism of plants against pathogens [18,19,20]. *TaGLP4* and *HvGLP4* silencing reduces basal resistance in wheat and barley, while overexpression of these two genes enhances resistance to *B. graminis* in both species [21]. After the *CchGLP* gene is silenced in pepper, pepper plants become more susceptible to PHYVV and PepGMV [19].

Melon (*Cucumis melo* L.) is an important horticultural crop in the world with important nutritional and economic values [22,23]. In the actual production process, melons are susceptible to low-temperature stress when planted in early spring and late autumn [24,25]. Low-temperature stress limits the growth and development of melon, reduces its yield and quality, and causes melon death in severe cases [26]. Although a large number of *GLPs* have been identified and analyzed in many plants, the *CmGLP* gene family system and expression patterns under low-temperature stress in melon have still been insufficiently reported. In the present study, we identified the members of the melon *GLP* gene family, analyzed their chromosomal location, gene structure, promoter cis-acting elements, evolutionary relationships, and replication events, and studied the binding of the transcription factor to the *CmGLP2-5* promoter via double luciferase assay. The *CmGLP* gene family relative expression levels in different melon tissues and *CmGLP* gene family expression patterns after low-temperature treatment were investigated by qRT-PCR. The results will provide a basis for the in-depth understanding of the evolution, function and low-temperature stress response of *CmGLPs* in melon.

## 2. Result

### 2.1. Identification, Chromosomal Location, and Physicochemical Properties Analysis of GLP Genes in Melon

A total of 36 candidate *GLPs* were obtained by searching ‘germin-like protein’ in the melon genome database Melon (DHL92). The structural domain analysis of 36 candidate melon GLP proteins led to the identification of 22 *CmGLP* genes with typical GLP structural domains (Table 1). The search results of ‘germin like protein’ using Ensembl plants are consistent with our final results. According to their position on the chromosome, the genes were named in the order of *CmGLP1-1*–*CmGLP12-2* (Figure 1). *CmGLPs* had a CDS sequence length of 498–747 and encoded a protein with an average length of 215 amino acids (aa), an average molecular weight of 22.9 kDa, and an isoelectric point (pI) of between 5.45–9.42. The prediction results showed that all members of the melon *GLP* gene family were hydrophobins, 20 members had signal peptides, and 6 members had transmembrane domains. Subcellular localization predictions showed that 11 family members were localized to the endoplasmic reticulum, and the remaining members were located in the extracellular, vacuolar, peroxisomal, and cytoplasmic.

*CmGLPs* were randomly distributed on eight chromosomes of melon, where the highest number of eight was observed on chromosome 8, followed by chromosome 2. The *CmGLP* gene family had a tandem duplication gene cluster containing five genes located on chromosome 8 (*CmGLP8-1*, *CmGLP8-2*, *CmGLP8-3*, *CmGLP8-4,* and *CmGLP8-5*). Two tandem duplication gene pairs were located on chromosome 2 (*CmGLP2-4*/*CmGLP2-5*) and chromosome 8 (*CmGLP8-6*/*CmGLP8-7*). In addition, seven pairs of segmental duplication genes were found within the *CmGLP* gene family (*CmGLP9-1*/*CmGLP12-2*, *CmGLP8-1*/*CmGLP12-2*, *CmGLP1-1*/*CmGLP10-1*, *CmGLP1-1*/*CmGLP2-1*, *CmGLP8-8*/*CmGLP12-2*, *CmGLP8-8*/*CmGLP12-1*, and *CmGLP8-8*/*CmGLP10-1*). Moreover, *CmGLP12-2* had fragment duplication with *CmGLP8-1* and *CmGLP8-2*, and these two genes had duplication events with multiple genes.

### 2.2. Analysis of Evolutionary Relationship between the GLP Proteins of Melon and Other Plants

To explore the evolutionary relationship of *GLPs* between melon and other species, we selected representative species and melon to construct phylogenetic trees (Figure 2). Arabidopsis is a representative plant of dicotyledons, rice belongs to monocotyledonous representative plants, and cucumber, like melon, belongs to Cucurbitaceae horticultural plants. We refer to the classification of the *GLP* gene family in cucumber by Liao et al. [27], and divide the *GLP* gene family into six subfamilies. In either melon, rice, or Arabidopsis, the *GLP* genes were randomly distributed in the five subfamilies (Group a–Group e). In addition to being distributed in these five subfamilies, the cucumber *GLP* gene family also had an additional Group f. Among the five subfamilies of the melon *GLP* gene family members, Group a had the largest number of members (nine members). The *GLPs* of Arabidopsis and rice also showed the same distribution.

### 2.3. Analysis of Protein Conserved Domains, Gene Structure, and Motifs of CmGLPs

The CDS sequences, UTR sequences, and intron positions of the *CmGLPs* members were mapped according to the results shown in the melon genome database (Figure 3A). Analysis of the gene structure revealed that each member of the *CmGLPs* gene family contains a typical Cupin_1 structural domain. Therefore, the structural domain of the Cupin of members of the melon *GLP* gene family has not diverged during evolution. A slight difference was observed in the number of introns among members of the *CmGLP* gene family in melon, except for *CmGLP12-2*, which contained four introns; all, other members contained only one intron or no intron. The introns containing gene family members *CmGLP2-1* and *CmGLP8-3* have an intron phase of 0; *CmGLP4-1*, *CmGLP8-1*, *CmGLP8-4*, and *CmGLP8-5* have an intron phase of 1; and *CmGLP2-2*, *CmGLP2-3*, *CmGLP5-1*, *CmGLP8-2*, and *CmGLP9-2* have an intron phase of 2. *CmGLP12-2* contains four introns, two of which have a phase of 2 and two of which have a phase of 1. In terms of conservatism: 0 phase was the highest, 2 phase was the lowest, and 1 phase was in the middle [28]. Intron phase is a conservative feature of eukaryotic gene structure, which is related to the evolution of introns in splice [29]. Meanwhile, some studies showed that the distribution of intron phases supports the hypothesis of the late appearance of introns [30].

Conserved motif analysis was performed on the protein sequences of the melon *CmGLP* gene family using MEME online software, and seven conserved motifs were obtained, namely motifs 1–7 (Figure 3B). A total of 17 genes in *CmGLPs* contain all seven conserved motifs. *CmGLP3-1* contains only two conserved motifs (motif 2 and motif 4) and differs significantly in number, type, and distribution from the conserved motifs of other genes. Among these conserved motifs, motif 2 is highly conserved, and all 22 *CmGLP* gene family members contain this motif.

### 2.4. Analysis of CmGLP Promoter cis-Acting Elements

Specific cis-elements in the 5′-upstream promoter (2.0 kb) region of 22 *CmGLPs* were analyzed using PlantCARE. In further analysis of the results, the cis-acting elements of the melon *CmGLPs* promoter could be divided into four main functional groups (Figure 4), hormone response elements (HREs), defense and stress response elements (DSREs), photoresponsive elements (LREs), and transcription factor regulatory elements. The HREs included auxin-responsive element, MeJA-responsiveness element, ethylene, and abscisic acid. The DSREs included low-temperature responsiveness elements, TC-rich, and drought responsive elements. The LREs included G-Box and I-Box. Moreover, the promoters of *CmGLPs* contain many binding sites for transcription factors, such as MYB, MYC, and W-box.

### 2.5. Relative Expression Analysis of CmGLP Gene Family in Different Tissue

The expression of the *CmGLP* gene in each main tissue of melon was analyzed by qRT-PCR, and the relative expression heat map was generated (Figure 5). All the members of the *CmGLP* gene family were expressed in the main tissues of melon, and the expression patterns differed in different tissues. The relative expression of *CmGLP9-2* was high in stems and seeds and low in fruits, indicating that it was preferentially expressed in stems and seeds. The relative expression levels of *CmGLP8-3* in stems and flowers were lower. Two genes, namely, *CmGLP3-1* and *CmGLP8-4,* had the highest relative expression in roots. *CmGLP2-2* had the highest relative expression in stems. Two genes, namely, *CmGLP1-1* and *CmGLP8-5*, had the highest relative expression in leaves. Five genes, namely, *CmGLP2-4*, *CmGLP2-5*, *CmGLP8-7*, *CmGLP10-1*, and *CmGLP12-1*, had the highest relative expression in flowers. Ten genes, namely, *CmGLP2-1*, *CmGLP4-1*, *CmGLP5-1*, *CmGLP8-1*, *CmGLP8-2*, *CmGLP8-3*, *CmGLP8-6*, *CmGLP9-1*, *CmGLP9-2*, and *CmGLP12-2*, had the highest relative expression in seeds. Two genes, namely, *CmGLP2-3* and *CmGLP8-8* had the highest relative expression in fruit. This result suggests that members of the *CmGLP* gene family can play important roles in various tissues of melon. Notably, nearly half of the *CmGLP* gene family members were highly expressed in seeds.

### 2.6. Expression Analysis of Melon CmGLP Gene Family in Melon Seedling Leaves under Low-Temperature Stress and Prediction of CmGLP2-5 Protein Structure

The expression levels of *CmGLP* gene family members in melon leaves were analyzed by qRT-PCR under low-temperature stress (Figure 6). The results showed that the *CmGLP* gene family was actively involved in the adaptation of melon to low-temperature stress. In comparison with CK, the 18 *CmGLPs* showed an up-regulated expression trend after low-temperature treatment. Among these 18 genes, the relative expression levels of 13 *CmGLPs*, namely, *CmGLP1-1*, *CmGLP2-1*, *CmGLP2-2*, *CmGLP2-4*, *CmGLP2-5*, *CmGLP3-1*, *CmGLP4-1*, *CmGLP8-2*, *CmGLP8-4*, *CmGLP8-5*, *CmGLP8-8*, *CmGLP10-1*, and *CmGLP12-2* reached the maximum at 48 h after low-temperature treatment. Notably, the relative expression of *CmGLP2-5* in the LT group was 157.13 times higher than that in the CK group after treatment at low temperature for 48 h. Among these 18 genes, the relative expression levels of the three *CmGLPs* (*CmGLP2-3*, *CmGLP8-1,* and *CmGLP8-6*) reached the maximum at 24 h after low-temperature treatment. The relative expression of *CmGLP8-6* in the LT group was 20.20 times higher than that in the CK group after treatment at low temperature for 24 h. Among these 18 genes, the relative expression levels of the two *CmGLPs* (*CmGLP9-2* and *CmGLP12-1*) reached the maximum at 12 h after low-temperature treatment. The relative expression levels of *CmGLP9-2* and *CmGLP12-1* in the LT group were 3.98 and 4.31 times that of the CK group, respectively, when treated at low temperature for 12 h. In comparison with CK, the relative expression levels of *CmGLP5-1* and *CmGLP8-7* showed a downward trend after low-temperature treatment. The relative expression levels of *CmGLP5-1* and *CmGLP8-7* in the LT group were 0.42 and 0.43 times that of the CK group when treated at low temperature for 12 h. The expression patterns of *CmGLP8-3* and *CmGLP9-1* under low-temperature stress were complex, which first decreased and then increased, and the lowest relative expression was observed at 12 h, whereas the highest expression was observed at 48 h. *CmGLP2-5* positively responded to low-temperature stress in melon seedling leaves. In order to increase the understanding of *CmGLP2-5*, we predicted its protein structure. The secondary structures of CmGLP2-5 protein included Random Coil, Extended Strand, and Alpha Helix, and the percentages of the three structures were 66.97%, 28.96%, and 4.07% (Appendix A). We also predicted the tertiary structural model of CmGLP2-5 protein (Appendix A). We used a Ramachandran plot to evaluate the quality of the tertiary structure model (Appendix A). The results showed that the amino acid residue conformation of the CmGLP2-5 protein had a good spatial conformation.

### 2.7. Dual Luciferase Assay

We selected the *CmGLP2-5* with the highest expression induced by low temperature. The cis-elements analysis of the promoter sequence of *CmGLP2-5* (2000 bp) found that its promoter sequence contained a large number of MYB-binding sites and W-box (Figure 4). Among them, W-box is considered to be the potential binding site of the WRKY transcription factor [31,32,33]. We cloned a fragment P in the *CmGLP2-5* promoter, which contains a W-box and three MYB-binding sites (Figure 7A). We recombined fragment P with the pGreen 0800-LUC vector to generate the reporter. Meanwhile, we recombined CmMYB23 and CmWRKY33 with pGreen 62-SK, respectively, to generate effectors (SK-CmMYB23 and SK-CmWRKY33). The reporter and effectors were co-transformed into tobacco leaves for dual luciferase assay. Quantitative dual luciferase assay demonstrated that CmMYB23 and CmWRKY33 regulate the expression of *CmGLP2-5* by interacting with fragment P (Figure 7C).

## 3. Discussion

GLP is a key class of enzymes, which play an important role in plant growth and development, defense against pathogens, and abiotic stress. The *GLP* gene family has been characterized in many plants, including *Oryza sativa* [34], *A. thaliana* [34], *Camellia sinensis* [16], *Triticum aestivum* L. [35], *Cucumis sativus* L. [27], and *Vitis vinifera* [36]. Although the melon’s full genome has been sequenced, the *GLP* gene family has not been identified by a genome-wide study in melon. This work involves the first genome-wide identification of the *CmGLP* gene family and analysis of *CmGLP* gene family characterization by using bioinformatics techniques.

### 3.1. Characteristics of the Melon GLP Gene Family

A total of 22 melon *GLP* genes were identified in this work (Table 1), and *CmGLPs* can be divided into five subfamilies according to their evolutionary relationships (Figure 2). Among them, Group a has nine members and is the largest subfamily in the *CmGLP* gene family. The number of members of the *GLP* gene family varies greatly in different species. For example, only seven *GLP*s [36] have been identified in grape, while 258 *GLPs* [35] have been identified in wheat. Gene duplication events such as tandem duplication and fragment duplication play an important role in the process of gene evolution, and gene duplication events often lead to the increase of gene family members [28]. The *CmGLP* gene family has a tandem repeat gene cluster, two tandem repeat gene pairs and seven pairs of segmental duplication genes (Figure 1). These gene duplication events drive the evolution of the *CmGLP* gene family to a certain extent. Notably, the *CmGLPs* on chromosome 8 are involved in the occurrence of multiple duplication events, which possibly explains the occurrence of most *CmGLP* gene family members on chromosome 8 in melon. Based on the analysis of the protein conserved domains, gene structure, and the motifs of *CmGLP* gene family, each member of the *CmGLP* gene family contains a typical Cupin_1 domain, most members have almost no difference in terms of the number of introns, and all members of the *CmGLP* gene family contain motif 2 (Figure 3). This indicates that members of the melon *GLP* gene family were relatively conserved in melon. We also found that the motifs of *CmGLPs* were only conserved in melon, and are quite different from the motifs of Arabidopsis, cucumber, and rice (Appendix A). The phylogenetic tree showed that *GLP* gene family members in Group a and Group c are located in different branches, and we found that Group f is a subfamily unique to cucumber. This indicates that *GLP* gene families of different species have diverged in evolution. The results of qRT-PCR show that the melon *GLP* gene family is expressed in the seeds, roots, stems, leaves, flowers, and fruits of melon, indicating the diversity of functions of the *CmGLP* gene family, which could play an important role in various life stages of melon (Figure 5). Studies on the *GLP* gene family of other plants show that the *GLP* gene family can be expressed in different tissues of the plant [37,38].

### 3.2. Potential Role of CmGLPs in Melon Response to Low-Temperature Stress

With the intensification of the greenhouse effect, extreme low temperatures continue to emerge and are constantly setting new records [39]. Low-temperature stress can cause serious damage to melon, resulting in the decline of melon yield and quality, thus restricting the development of the melon industry [40,41]. *GLPs* play an active role in plants coping with abiotic stresses, including low-temperature stress [42,43,44]. Based on the analysis of the cis-acting elements of the promoters of the melon *GLP* gene family, the promoters of the members of the melon *GLP* gene family contained a large number of DSREs, including low-temperature responsiveness elements (Figure 4). This finding further indicates that the melon *GLP* gene family could be regulated by low temperature. However, we found few studies on *CmGLPs* in melon under low-temperature stress. Therefore, the study of *CmGLPs* as an important low-temperature regulator is expected to be an interesting entry point for exploring the relationship between melon and low-temperature stress. qRT-PCR results in the leaves of melon seedlings showed that the relative expression levels of 18 *CmGLP* gene family members showed an upward trend after low-temperature stress, and the relative expression levels of two *CmGLP* gene family members showed a downward trend after low-temperature stress, and the relative expression levels of two *CmGLP* gene family member initially decreased and then increased (Figure 6). This finding suggests that most members of the *CmGLP* gene family actively respond to low-temperature stress by up-regulating their expression. The different expression patterns of different *CmGLPs* under low-temperature stress indicate that the *CmGLPs* involved in the low-temperature regulation mechanism of melon are complex. Among the upregulated genes after low-temperature stress, *CmGLP2-5* had the largest change. Under low-temperature stress for 48h, the relative expression level of *CmGLP2-5* in the LT group was 157.13 times that of the CK group at the same time point. The *CmGLP2-5* may play a key role in the response of melon seedlings to low-temperature stress. AS similar research result was shown in *C. plantagineum* after treatment with mannitol and exogenous ABA, and the *CpGLP1* expression significantly increased, thus supporting the participation of *CpGLP1* in adapting to drought stress processes [45]. *GmGLP7* was transformed into Arabidopsis, and the transgenic Arabidopsis exhibited good salt tolerance and was sensitive to ABA treatment [43]. Therefore, low-temperature resistance in melon can be improved by transforming *CmGLP* through overexpression, gene editing, and other techniques. Functional proteins are often regulated by upstream transcription factors when they function [46,47,48]. Based on the analysis of the promoter elements of *CmGLP2-5*, the promoter of *CmGLP2-5* contains a large number of transcription factor regulatory elements, including the MYB binding site and W-box (Figure 4 and Figure 7A). *MdMYB23* enhances low-temperature tolerance in apples by accumulating proanthocyanidin [49]. *VaWRKY33* enhances the low-temperature tolerance of grapes by responding to ethylene [50]. In the present study, we cloned the homologous genes *CmMYB23* and *CmWRKY33* in melon and demonstrated by dual luciferase assay that *CmMYB23* and *CmWRKY33* interact with fragment P to regulate the expression of *CmGLP2-5*. The result provided further insights into the regulatory patterns of the melon *GLP* gene family members.

## 4. Materials and Methods

The flowchart of methodology is shown in Appendix A.

### 4.1. Plant Material and Stress Treatments

Melon (cv ‘*Qianyu No. 6*’) seedlings were cultivated as previously described [23,28] and grown in a controlled environment greenhouse. The culture conditions for melon seedlings were 25 °C day/18 °C night, 65% relative humidity, 15,000 LX light intensity, and a 12 h day/12 h night. The experiments began when the third true leaves were completely expanded.

Half of the melon seedlings were kept under normal temperature (25 °C/18 °C, day/night), and the remaining melon seedlings were exposed to low temperature (4 °C/4 °C, day/night) with the same air humidity and light condition as the normal temperature melon seedlings. The second true leaf of the melon seedlings was sampled at 0, 12, 24, and 48 h after temperature treatment. All melon seedling samples were immediately frozen in liquid nitrogen and stored at −80 °C for RNA extraction. Three biologicals and three technical replicates were used in the experiment.

### 4.2. Identification of Melon GLP Gene Family Members

For the identification of *GLP* in melon, we used the following strategy: First, the candidate sequence was obtained by searching ‘germin-like protein’ in melon genome database (http://cucurbitgenomics.org/organism/18, accessed on 12 April 2021) DHL92 [51]. Then, the NCBI Conserved Domain database (CDD, https://www.ncbi.nlm.nih.gov/Structure/cdd/wrpsb.cgi, accessed on 12 April 2021) was employed to identify candidate sequences. To ensure the accuracy of the genetic screening, we also used the Ensembl Plants (http://plants.ensembl.org/index.html, accessed on 18 July 2022) to search for ‘germin-like protein’ [52]. Finally, on the basis of the search results to determine the melon *GLP* member, the candidate sequence with a Cupin_1 domain was determined to be a melon *GLP* [34].

### 4.3. Chromosome Localization and Gene Duplication of Melon CmGLP Gene Family

The length of each melon chromosome and the position of each member of the *CmGLP* gene family were identified through the melon genome database (DHL92). Duplication events of *CmGLP* gene family members were analyzed using MCscanX software [53], and only ≥3 Mb duplicate fragments were selected as background events [54]. Combined with the chromosomal location information of *CmGLP* gene family members, Circos software was used to map the tandem replication.

### 4.4. Analysis of Melon CmGLP Gene Family Characteristics

The characteristics of each *CmGLP* gene were analyzed using the online analysis software of ExPASY (http://www.expasy.org, accessed on 19 June 2021) [55]. GenScript (https://www.genscript.com/, accessed on 22 June 2021) was employed to predict the subcellular location of each *CmGLP* gene. Transmembrane domains were predicted using TMHMM online software (https://services.healthtech.dtu.dk/service.php?TMHMM-2.0, accessed on 18 July 2022) [56]. Signal peptides were predicted using the Signal peptides online software (https://services.healthtech.dtu.dk/service.php?SignalP-4.1, accessed on 18 July 2022) [57].

### 4.5. Phylogenetic, Gene Structure, Multiple Sequence Alignment, Promoter cis-Acting Elements Analysis of Melon GLP Gene Family

The protein sequences of the *GLP* gene family of *Arabidopsis thaliana* (https://www.arabidopsis.org/, accessed on 1 July 2021) [58], *Oryza sativa* L. (http://rice.uga.edu/, accessed on 2 July 2021) [59], and *Cucumis sativus* L. (http://www.cucurbitgenomics.org/organism/2, accessed on 18 July 2022) [60] were downloaded. A total of 43 rice GLP proteins, 32 Arabidopsis GLP proteins, and 38 cucumber GLP proteins were used. The gene sequence numbers correspond to the proteins listed in Appendix A. MEGA 5.0 software was employed to analyze the above sequences and melon CmGLP protein sequence (ClustalW analysis) and finally draw the evolutionary tree [61]. The conserved domains of melon *GLP* gene family proteins were analyzed using the NCBI conserved domain database (https://www.ncbi.nlm.nih.gov/, accessed on 7 July 2021). IBS software [62] was applied to map the gene structure of *CmGLP* genes. The conserved motifs of melon *GLP* gene family members in melon were analyzed by MEME online website (http://meme-suite.org/tools/meme, accessed on 16 July 2021) [63]. The promoter region 2000bp upstream of the translation initiation site (ATG) of *CmGLPs* was selected for cis-acting regulatory elementanalysis prediction by the online analysis software PlantCARE (https://bioinformatics.psb.ugent.be/webtools/plantcare/html/, accessed on 6 August 2021) [64].

### 4.6. Relative Expression Analysis of CmGLP Gene Family in Different Tissue

All major tissues of a healthy growing melon, including seeds, roots, stems, leaves, flowers, and ripe fruits, were washed and quickly placed in liquid nitrogen. Twenty plump melon seeds were selected as samples. When the fourth leaf is fully unfolded, the third leaf and the root system are sampled, while the stem between the third and fourth leaf is sampled. Completely open female and male flowers were selected, and the samples were mixed and sampled according to the quality of 1:1. The seeds of fully ripe melon fruits were first removed, and then the fruits were pulped and sampled. Except for seeds, other tissues were selected from 3 independent plants for sampling. All samples were stored at −80 °C for RNA extraction.

### 4.7. RNA Extraction and qRT-PCR Analyses

Total RNA was extracted following the method of Liu et al. [65]. Total RNA was reverse-transcribed into cDNA following the method of Zhang et al. [23]. qRT-PCR analysis was performed following the method of Liu et al. [65]. Gene-specific primers and *actin7* are listed in Appendix A. *Actin7* expression was used as an internal standard, and the expression levels of other genes were normalized using the comparative Ct method [45].

### 4.8. CmGLP2-5 Protein Structure Prediction

We used the ALIGNSEC online website (https://npsa-prabi.ibcp.fr/cgi-bin/npsa_automat.pl?page=npsa_gor4.html, accessed on 17 July 2022) to predict the secondary structure of CmGLP2-5 protein [66]. We used the SWISS-MODEL online website (https://swissmodel.expasy.org/interactive, accessed on 17 July 2022) to predict the tertiary structure of the CmGLP2-5 protein [67]. We plotted the Ramachandran plot using the PDBsum online software (http://www.ebi.ac.uk/thornton-srv/databases/pdbsum/Generate.html, accessed on 17 July 2022) to assess the plausibility of the tertiary structure of the CmGLP2-5 protein [68].

### 4.9. Dual Luciferase Assay

The CDSs of *CmMYB23* and *CmWRKY33* were inserted into the pGreen 62-SK vector, respectively, to generate two effectors (SK-CmMYB23 and SK-CmWRKY33). The promoter fragment of *CmGLP2-5* was recombined to the pGreen 0800-LUC vector to generate the reporter. The constructs were co-transformed into tobacco seedlings as described in a previous report [69]. The transformed tobacco seedlings were cultured in the dark for 2 days to conduct a dual luciferase assay. Luciferase activities were measured with *TransDetect*^®^ Double-Luciferase Reporter Assay Kit (TransGen Biotech, Beijing, China) according to the manufacturer’s instruction. The primers used for Dual luciferase assay are shown in Appendix A.

### 4.10. Statistical Analysis

This experiment used three biologicals and three technical replicates. Statistical analysis of the bioassays was performed using the SAS software version 8.0 (SAS Institute, Cary, NC, USA) through the Tukey’s test at a level of *p* < 0.05.

## 5. Conclusions

In the present study, 22 *CmGLPs* were identified in melon, and the melon *GLP* gene family was analyzed in detail in terms of chromosomal location, physicochemical properties, phylogenetic relationships, protein conserved domains, gene structure, motifs, promoter cis-acting elements, gene duplication, expression levels in various tissues, and low-temperature expression pattern analysis. A large number of gene duplication events promoted the evolution of the melon *GLP* gene family to a certain extent. The melon *GLP* gene family is relatively conserved in the evolutionary process with limited functional differentiation. The results of qRT-PCR showed that the melon *GLP* gene family was expressed in various tissues of melon and actively participated in the low-temperature response. *CmGLP2-5* played an important role in melon low-temperature responses. The dual luciferase assay showed that *CmMYB23* and *CmWRKY33* could regulate the expression of *CmGLP2-5* by acting on the promoter fragment. The genome-wide investigation of the *CmGLP* provides a good framework for functional assessments of *GLPs* in melon. Taken together, this study not only provides comprehensive information about the melon *GLP* gene family but also lays the foundation for genetic modification technologies to improve melon low-temperature tolerance.

## Figures and Tables

**Figure 1 ijms-23-08190-f001:**
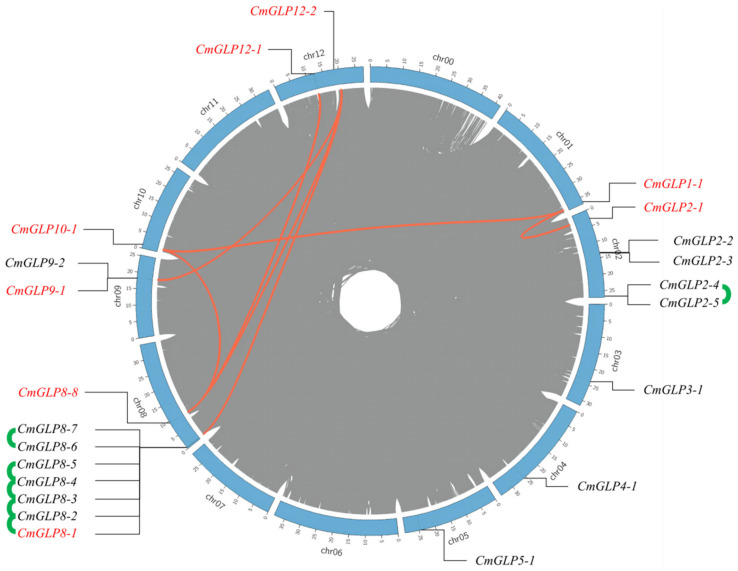
Chromosomal localization and collinear distribution of melon *GLP* genes. The gray line indicates the collinear block between the melon genome; the green line indicates the tandem duplication gene pairs of the *CmGLP*, gene names marked in red and connected with red lines indicate segmented duplication genes.

**Figure 2 ijms-23-08190-f002:**
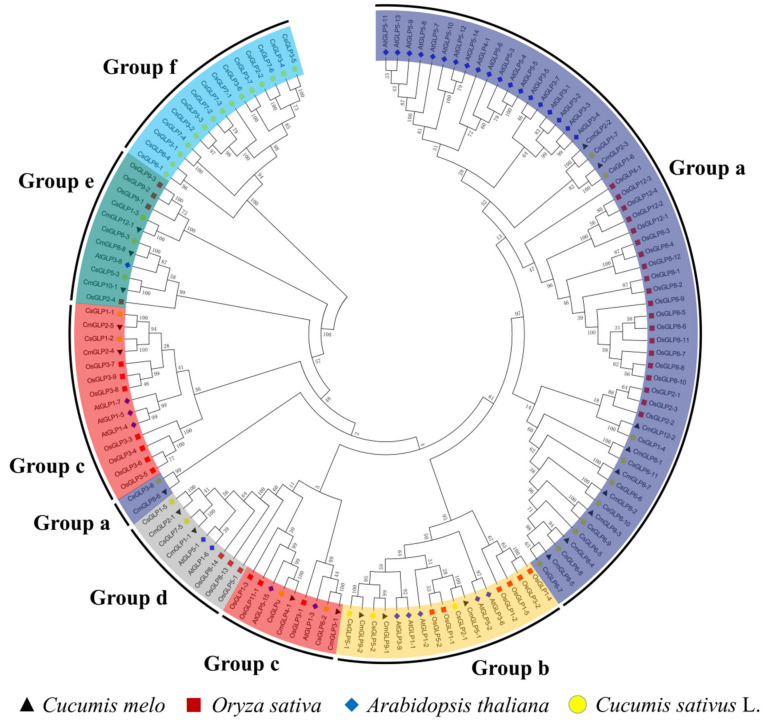
Phylogenetic tree of *GLP*s from melon (Cm), Arabidopsis thaliana (At), rice (Os), and cucumber (Cs). To distinguish different subfamilies by the color of evolutionary branches.

**Figure 3 ijms-23-08190-f003:**
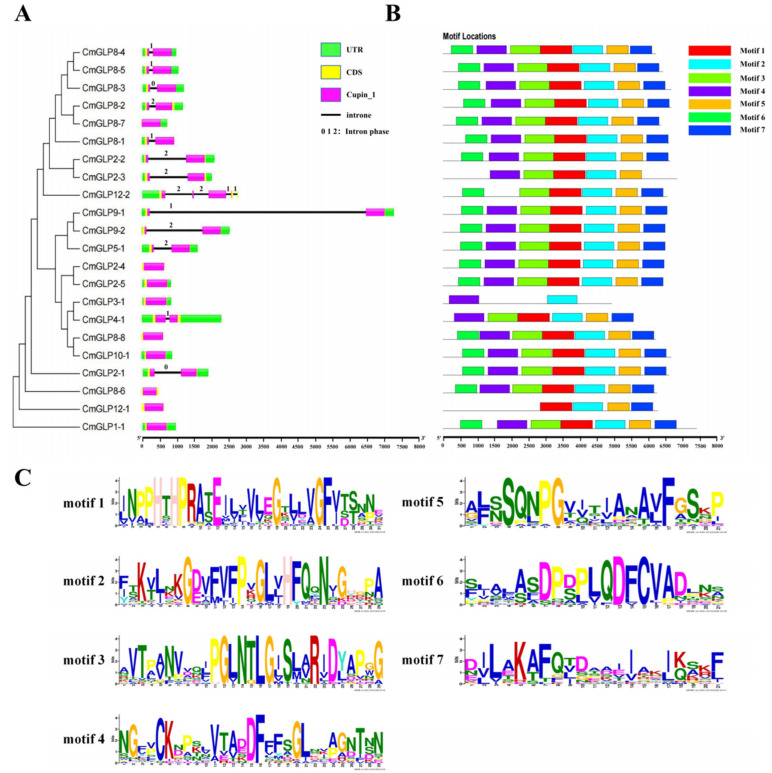
Gene structure (**A**) of *CmGLPs* and the conserved motifs (**B**,**C**) in their encoded proteins. (**A**) Gene structures of *CmGLP* genes in melon. UTR, CDS, Cupin_1 and introns were represented by green boxes, orange boxes, pink boxes and black lines, respectively. The size of CDS and introns can be estimated using the ratio of the bottom. The numbers of 0, 1, and 2 represent the splicing phase of intron. (**B**) Schematic representation of the conserved motifs predicted in the CmGLP proteins. The different colours represent different conserved motifs. The length of the amino acid was inferred by ruler at bottom. (**C**) Length and amino acid species of the seven conserved motifs of the CmGLPs protein sequence. The size of letters represents the frequency of amino acid occurrence.

**Figure 4 ijms-23-08190-f004:**
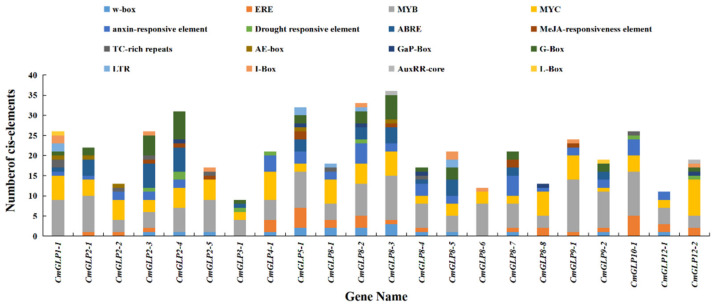
Predicted cis-elements in *CmGLP* promoters of melon. Different cis-elements were identified and plotted against a bar diagram. The abundance of different regulatory elements on each of the promoters is shown in different colors.

**Figure 5 ijms-23-08190-f005:**
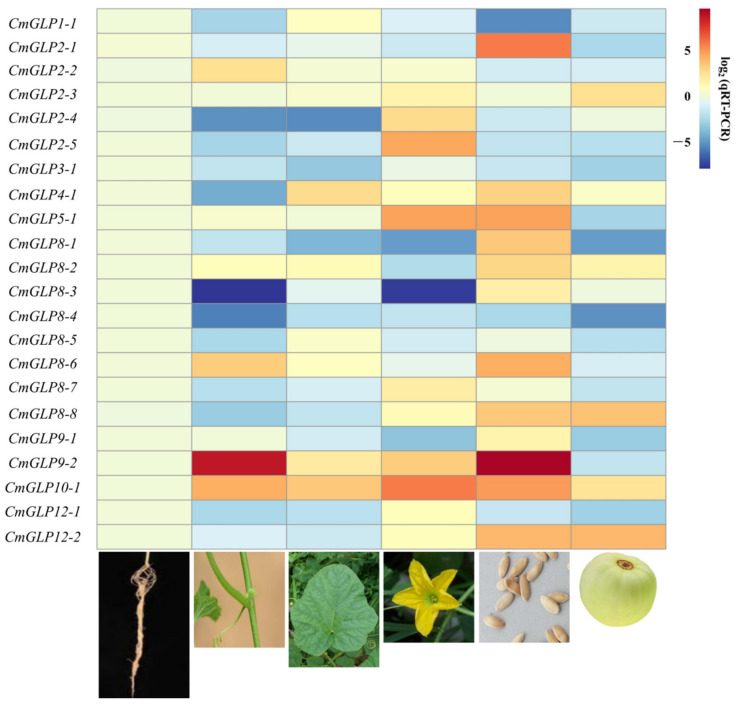
Heat map of relative expression differences of *GLP* genes in different tissues of melon based on qRT-PCR analysis. Each column represents the relative expression levels of *CmGLP* genes in roots, stems, leaves, flowers, seeds. The expression level of each gene in the root was used as a control, with log2 processing on the qRT-PCR value. Blue indicates lower expression levels and red indicates higher expression levels.

**Figure 6 ijms-23-08190-f006:**
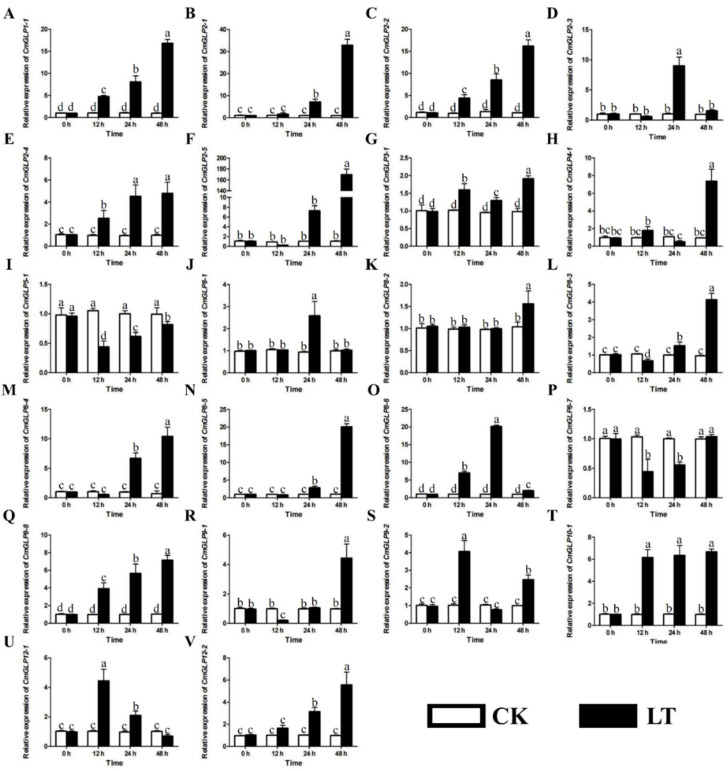
Relative expression of melon *GLP* gene family members in melon leaves under low-temperature (LT) stress. (**A**–**V**) Relative expression levels of melon GLP gene family members in leaves of melon seedlings after low-temperature stress. Data are expressed as the mean ± standard error of three independent biological replicates. Different letters above the bars indicate a significant difference at *p* < 0.05.

**Figure 7 ijms-23-08190-f007:**
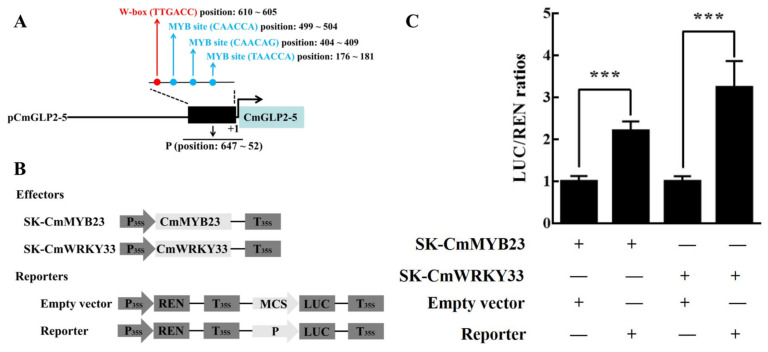
Dual luciferase assay *CmMYB23* and *CmWRKY33* binding to *CmGLP2-5* promoter. (**A**) Schematic diagram of *CmGLP2-5* promoter, black rectangle indicates the W-box and MYB-binding elements in the partial promoter fragment (P). (**B**) Schematic diagrams of effector and reporter constructs used for dual-LUC transient assay. (**C**) The various combinations of vectors were performed on Dual-LUC transient expression assays. Asterisks indicate that the value is significantly different from that of the control (*** *p* < 0.001).

**Table 1 ijms-23-08190-t001:** The information on the *GLP* gene family in melon.

No.	Gene Name	Gene ID	Length (bp)			*M*_w_ (kDa)	Exons	pI	Singal Peptide	Trans- Membrane Domain	Protein (aa)	GRAVY	Localization
			Gene	cDNA	CDS								
1	*CmGLP1-1*	MELO3C024356	966	966	627	21.37	1	6.02	Yes	0	208	0.617	vacuolar
2	*CmGLP2-1*	MELO3C015515	1906	1112	648	22.42	2	7.70	Yes	1	215	0.506	endoplasmic reticulum
3	*CmGLP2-2*	MELO3C010114	2117	988	672	24.10	2	6.27	Yes	0	223	0.271	endoplasmic reticulum
4	*CmGLP2-3*	MELO3C010121	2032	895	672	24.03	2	7.82	Yes	0	223	0.355	endoplasmic reticulum
5	*CmGLP2-4*	MELO3C017139	642	642	642	23.05	1	8.54	Yes	0	213	0.448	endoplasmic reticulum
6	*CmGLP2-5*	MELO3C017138	821	821	666	23.59	1	8.82	Yes	1	221	0.386	extracellular
7	*CmGLP3-1*	MELO3C019788	831	831	669	24.05	1	8.71	Yes	0	222	0.038	endoplasmic reticulum
8	*CmGLP4-1*	MELO3C026976	2329	2208	690	25.08	2	6.28	Yes	1	229	0.212	extracellular
9	*CmGLP5-1*	MELO3C004143	1620	1081	660	23.08	2	9.42	Yes	0	219	0.204	vacuolar
10	*CmGLP8-1*	MELO3C007061	940	738	660	23.33	2	7.77	Yes	0	219	0.255	endoplasmic reticulum
11	*CmGLP8-2*	MELO3C007062	1178	976	654	22.86	2	5.84	Yes	0	217	0.406	peroxisomal
12	*CmGLP8-3*	MELO3C007063	1198	984	654	23.27	2	7.78	Yes	1	217	0.297	endoplasmic reticulum
13	*CmGLP8-4*	MELO3C007064	972	851	651	22.80	2	5.84	Yes	0	216	0.379	peroxisomal
14	*CmGLP8-5*	MELO3C007065	1045	922	648	22.60	2	6.03	Yes	0	215	0.354	endoplasmic reticulum
15	*CmGLP8-6*	MELO3C007067	498	498	498	18.00	1	5.65	No	0	165	0.076	cytoplasmic
16	*CmGLP8-7*	MELO3C032878	755	755	561	19.73	1	6.90	No	0	186	0.319	cytoplasmic
17	*CmGLP8-8*	MELO3C024541	627	627	627	22.54	1	8.91	Yes	0	208	0.114	cytoplasmic
18	*CmGLP9-1*	MELO3C005184	7426	1028	672	23.33	2	8.72	Yes	0	223	0.130	endoplasmic reticulum
19	*CmGLP9-2*	MELO3C005186	2576	924	666	23.52	2	5.50	Yes	1	221	0.260	extracellular
20	*CmGLP10-1*	MELO3C012352	885	885	630	22.53	1	6.37	Yes	0	209	0.158	extracellular
21	*CmGLP12-1*	MELO3C004688	633	633	633	22.62	1	5.45	Yes	2	210	0.331	endoplasmic reticulum
22	*CmGLP12-2*	MELO3C002829	2833	1246	747	26.59	1	8.50	Yes	0	248	0.168	endoplasmic reticulum

## Data Availability

All the raw data, as well as gene annotations can be found in the Solanaceae Genomics Network (https://solgenomics.net/, accessed on 25 July 2022). All other data are available from the corresponding author upon reasonable request.

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
