# Peer review of "Genome-Wide Identification, Characterization, and Expression Analysis Related to Low-Temperature Stress of the CmGLP Gene Family in Cucumis melo L."

_ijms, 2022, doi:10.3390/ijms23158190_

Round 1
Reviewer 1 Report
The article describes the analysis of a group of germin-like proteins in melon. The study contains extensive genomic analysis and some qRT-PCR analysis of gene expression in different tissues and in seedlings in response to cold stress conditions.
Prediction analysis based on the protein sequences of the 22 genes with GLP domains is described in Table 1. It is unclear how the hydrophobicity and subcellular localization of these proteins are related to each other (for example, hydrophobic proteins are expected to be located in the membranes) and how they compare to the GLP proteins characterized in other species, such as Arabidopsis and rice.
In addition, do all of these 22 genes have homologues in Arabidopsis and rice? Are there other genes that could be identified in melon with the BLAST searchers using the GLP proteins in other species?
The phylogenetic relationships of the 22 identified genes are shown in Figure 2 (is there Figure 1 in the article ?) It is unclear why some subfamilies have only the numbers and others are called by the plant taxa they are found in. Maybe adding some of the representative GLP-like sequences from other species would be helpful?
Conserved motifs are shown in Figure 3. Are these motifs conserved in Arabidopsis and rice GLP sequences, too? Or are they only conserved in melon? What is the importance of intron phases?
The promotor motifs are shown in Figure 4. This part of the paper would benefit from distinguishing between the motifs that are experimentally shown to have a particular function, motifs that are shown to be present in genes with particular functions, etc... If different GLP proteins could have different functions (as discussed in the introduction), they would be expected to have different motifs.
Gene expression analysis: there is no description of how different tissues were collected for this analysis. The authors need to provide better description about how the values for Figure 5 were calculated. In addition, they used actin as a reference gene. Is it experimentally verified that the expression of actin remains stable in different tissues in melon?
Figure 6: The results here are interesting, but the authors should clarify more here that the cold stress investigated here is only in seedling leaves, where most of the GLP genes were expressed at relatively low levels.
Figure 7 (Luciferase essay) - this portion and Figure 7 are very confusing. It seems that the results show that a particular portion of the promoter drives the expression of the reporter gene. What is the importance of this data? Isn't this motif shown to control gene expression in other species? Additional expression studies under different conditions or comparative analysis of different areas of the promoters of GLP genes in melon would strengthen this part of the study.
Author Response
Response to Reviewer 1 Comments
Point 1: Prediction analysis based on the protein sequences of the 22 genes with GLP domains is described in Table 1. It is unclear how the hydrophobicity and subcellular localization of these proteins are related to each other (for example, hydrophobic proteins are expected to be located in the membranes) and how they compare to the GLP proteins characterized in other species, such as Arabidopsis and rice.
Response 1: Thanks for your comment. As you said, there is currently no evidence about the directly relation between the hydrophobicity and subcellular localization of a protein. Membrane proteins are mainly spherical proteins embedded between lipid bilayers. Their hydrophilic ends are exposed on the membrane surface, and their hydrophobic ends are embedded in the lipid bilayer to connect with the hydrophobic part of lipid molecules. Some cell membrane proteins run through the whole lipid bilayer. Meanwhile, some proteins did not embed in the lipid bilayer but only attached to the inner surface of the lipid bilayer (Vereb et al., 2003; Raghunathan and Kenworthy, 2018). The location of the protein in the cell may be related to the signal peptide, but the mechanism of the specific effect of the signal peptide on the subcellular localization of the protein is still unclear (Nakai 2000; Savojardo et al., 2018). We put the results of hydrophobicity and subcellular localization prediction here to show the information of melon GLP gene family, which is not necessarily related to each other. As your comments, we added the prediction of the transmembrane domain and signal peptide of the melon GLP gene family in Table 1 to show more information about the melon GLP gene family, thereby increasing the appeal of the manuscript (Table 1, Line: 201). In addition, we can use DNAMAN software to compare the sequence differences between melon GLP and GLP of other species. If it is a difference in basic information, such as the prediction of subcellular localization, we can compare by looking for published articles (Li et al., 2016), or we can predict the GLP of other species using the same method for comparison. Thanks again for your comment.
Li, L.; Xu, X.; Chen, C.; et al. Genome-wide characterization and expression analysis of the germin-like protein family in rice and Arabidopsis. Int. J. Mol. Sci. 2016. 17, 1622.
Nakai K. Protein sorting signals and prediction of subcellular localization. Adv. Protein Chem. 2000. 54, 277-344.
Raghunathan, K.; Kenworthy, A.K. Dynamic pattern generation in cell membranes: Current insights into membrane organization. BBA-BIOMEMBRANES. 2018. 1860(10).
Savojardo, C.; Martelli, P.L.; Fariselli, P.; et al. BUSCA: an integrative web server to predict subcellular localization of proteins. Nucleic Acids Res. 2018. 46, W459-W466.
Vereb, G.; Szöllősi, J.; Matko, J; et al. Dynamic, yet structured: the cell membrane three decades after the Singer–Nicolson model. PNAS. 2003. 100, 8053-8058.
Point 2: In addition, do all of these 22 genes have homologues in Arabidopsis and rice? Are there other genes that could be identified in melon with the BLAST searchers using the GLP proteins in other species?
Response 2: Thanks for your comment. After our comparison and analysis, all 22 members of the melon GLP gene family can find homologous genes in Arabidopsis and cucumber. However, CmGLP2-4 and CmGLP4-1 did not find homologous genes in rice. This may be due to the large genetic difference between melon (dicotyledonous plants) and rice (monocotyledonous plant). According to the results of the phylogenetic tree, we also found that GLP gene families of different species have diverged in evolution (Line: 393). We used Arabidopsis, rice and cucumber GLPs to BLAST search the melon genome database and didn’t finnd new melon GLP. To ensure the accuracy of the manuscript, we searched again for "Germin-like protein" using the Ensembl Plants database. The results of the search again are consistent with our previous results, still 22 melon GLP genes. Above work had been supplemented in the manuscript (Lines: 113, 191).
Point 3: The phylogenetic relationships of the 22 identified genes are shown in Figure 2 (is there Figure 1 in the article ?) It is unclear why some subfamilies have only the numbers and others are called by the plant taxa they are found in. Maybe adding some of the representative GLP-like sequences from other species would be helpful?
Response 3: Thanks for your good suggestion. At present, there is no uniform standard for the classification of GLP gene family, and different researchers have different names for GLP gene family (Li et al., 2016; Fan et al., 2020; Liao et al., 2021). Some researchers use the species under study to name newly discovered subfamilies (Dunwell et al., 2008). In order to better present our findings, we reclassified the GLP gene family according to the taxonomy of GLP gene family in cucumber (Liao et al., 2021). We also added the Cucumber (belongs to the horticultural plant of Cucurbitaceae) GLP protein sequence to the phylogenetic tree according to your opinion, and reconstructed the phylogenetic tree (Fig. 2, Line: 231). In addition, there is figure 1 in our manuscript (Line: 214).
Dunwell, J. M.; Gibbings, J. G.; Mahmood, T.; et al. Germin and germin-like proteins: evolution, structure, and function. Crit. Rev. plant Sci. 2008. 27, 342-375.
Fan, LIU.; Na, T.; Xueli, SUN.; et al. Genome-wide Identification and Expression Analysis of Banana GLP Gene Family. Acta Horticulturae Sinica. 2020. 47, 1930.
Li, L.; Xu, X.; Chen, C.; et al. Genome-wide characterization and expression analysis of the germin-like protein family in rice and Arabidopsis. Int. J. Mol. Sci. 2016. 17, 1622.
Liao, L.; Hu, Z.; Liu, S.; et al. Characterization of Germin-like Proteins (GLPs) and Their Expression in Response to Abiotic and Biotic Stresses in Cucumber. Horticulturae. 2021. 7, 412.
Point 4: Conserved motifs are shown in Figure 3. Are these motifs conserved in Arabidopsis and rice GLP sequences, too? Or are they only conserved in melon? What is the importance of intron phases? The promotor motifs are shown in Figure 4. This part of the paper would benefit from distinguishing between the motifs that are experimentally shown to have a particular function, motifs that are shown to be present in genes with particular functions, etc...If different GLP proteins could have different functions (as discussed in the introduction), they would be expected to have different motifs.
Response 4: Thanks for your questions and comments. After analysis and comparison, we found that the motifs of melon GLPs only conserved in melon and are quite different from those of Arabidopsis, cucumber, and rice. We have supplemented this new finding in the discussion section (Line: 389). Intron phase is a conservative feature of eukaryotic gene structure, which related to the evolution of introns in splice (Yang et al., 2019). Meanwhile, some studies shown that the distribution of intron phases supports the hypothesis of late appearance of introns (Long et al., 1995). These results help to explore the origin of introns. In view of your question, we have added some clarification about the intron phases in section 3.3 of the manuscript (Lines: 247-250).
Long, M., Rosenberg, C., Gilbert, W. Intron phase correlations and the evolution of the intron/exon structure of genes. PNAS. 1995. 92, 12495-12499.
Yang, G.; Lu, H.; Wang, L.; et al. Genome-wide identification and transcriptional expression of the METTL21C gene family in chicken. Genes. 2019. 10, 628.
Point 5: Gene expression analysis: there is no description of how different tissues were collected for this analysis. The authors need to provide better description about how the values for Figure 5 were calculated. In addition, they used actin as a reference gene. Is it experimentally verified that the expression of actin remains stable in different tissues in melon?
Response 5: Thank you very much for your suggestion. We have detail re-described melon tissue sampling process in section 2.6. Regarding the question of how to calculate in Figure 5, we have made additions in the legend (Line: 303). Our previous study showed that CmActin-7 as a reference gene could be stably expressed in different tissues of melon (Wang et al., 2020; Zhang et al., 2020a; Zhang et al., 2020b).
Wang, J.; Zhang, Z.; Wu, J.; et al. Genome-wide identification, characterization, and expression analysis related to autotoxicity of the GST gene family in Cucumis melo L. Plant Physiol. Bioch. 2020. 155, 59-69.
Zhang, Z.; Zhang, Z.; Han, X.; et al. Specific response mechanism to autotoxicity in melon (Cucumis melo L.) root revealed by physiological analyses combined with transcriptome profiling. Ecotox. Environ. Safe. 2020a. 200, 110779.
Zhang, Z.; Fan, J.; Wu, J.; et al. Alleviating effect of silicon on melon seed germination under autotoxicity stress. Ecotox. Environ. Safe. 2020b. 188, 109901.
Point 6: The results here are interesting, but the authors should clarify more here that the cold stress investigated here is only in seedling leaves, where most of the GLP genes were expressed at relatively low levels.
Response 6: Thanks a lot for your suggestion. We have supplemented some descriptions of 'studies in seedling leaves" in the results (Lines: 306, 308, 340) and discussion (Line: 411) sections of the manuscript. Low-temperature stress increased the expression of most CmGLPs in the leaves of melon seedlings.
Point 7: Figure 7 (Luciferase essay) - this portion and Figure 7 are very confusing. It seems that the results show that a particular portion of the promoter drive the expression of the reporter gene. What is the importance of this data? Isn't this motif shown to control gene expression in other species? Additional expression studies under different conditions or comparative analysis of different areas of the promoters of GLP genes in melon would strengthen this part of the study.
Response 7: Thanks for your suggestion. In order to facilitate understanding, we have followed your suggestion and added some descriptive sentences in section 3.7 (Lines: 347, 349-353). Through the analysis of gene expression in the leaves of melon seedlings under low-temperature stress, we screened out a gene (CmGLP2-5) that actively responds to low-temperature stress. There are many potential binding sites for transcription factors on the promoter sequence of CmGLP2-5, so we wanted to explore which transcription factors could regulate the gene by binding to the promoter of CmGLP2-5. The dual luciferase assay is one of the important experiments to verify whether transcription factors regulate functional proteins by binding to promoters (Wang et al., 2022; Yang et al., 2022; Zhang et al., 2022). The reason why CmMYB23 and CmWRKY33 were selected for validation is that these two transfactors have been shown to be involved in low temperature regulation (The Discussion section is explained. Lines: 433-435). Figure 7A shows the specific locations of our selected fragments in the CmGLP2-5 promoter. This fragment contains potential binding sites for MYB and WRKY transcription factors. Figure 7B shows the vector information required for dual luciferase assay. Figure 7C shows the results of the dual luciferase assay. All the presentations refer to previous studies (Wang et al., 2022; Yang et al., 2022; Zhang et al., 2022). The dual luciferase assay showed that CmMYB23 and CmWRKY33 could regulate the expression of CmGLP2-5 by binding to the promoter of CmGLP2-5. Thank you again for your valuable comments.
Wang, F.; Wang, X.; Zhang, Y.; et al. SlFHY3 and SlHY5 act compliantly to enhance cold tolerance through the integration of myo‐inositol and light signaling in tomato. New Phytol. 2022. 233, 2127-2143.
Yang, X.; Luo, Y.; Bai, H.; et al. DgMYB2 improves cold resistance in chrysanthemum by directly targeting DgGPX1. Hortic. Res. 2022, 9.
Zhang, Y.; Ming, R.; Khan.; M, et al. ERF9 of Poncirus trifoliata (L.) Raf. undergoes feedback regulation by ethylene and modulates cold tolerance via regulating a glutathione S‐transferase U17 gene. Plant Biotechnol. J. 2022. 20, 183.
Reviewer 2 Report
In this manuscript, authors have screened the 22 Germin-like protein (GLP) in melon genome with utilizing sequencing data available in NCBI and annotated them using open-source bioinformatics tools followed by stress specific validation using qRT-PCR. Based on the phylogeny and motif patterns, these CmGLPs have been classified in seven classes.
Manuscript is good and belongs to the scope of journal but need major revisions before acceptance.
Following are my comments and suggestions for authors to improve the quality of MS.
1. Provide the flowchart of methodology to understand concept in more attractive manner.
2. Why authors have only searched the NCBI CDD for mining of genes? Why you have not searched the Ensembl Plants ? Try to explore the Ensembl Plants for more genes.
3. Extensive information are required in "section 2.2".
4. Cite the all tools and databases in significant manner only URLs are not enough.
5. In section 2.5, define the how many genes you have derived from Arabidopsis and rice for phylogeny analysis? Authors are advised to collected more genes from evolutionary closed species to conduct extensive phylogeny analysis.
6. Authors are advised to conduct the target miRNAs analysis.
7. Evaluate the gene specific SSR markers potential of identified genes.
8. Perform the protein structure modeling of representative proteins followed by structure evaluation using Ramachandran plot.
Author Response
Point 1: Provide the flowchart of methodology to understand concept in more attractive manner.
Response 1: Thank you very much for your suggestion. We have added the flowchart of methodology to section 2 (Line: 94).
Point 2: Why authors have only searched the NCBI CDD for mining of genes? Why you have not searched the Ensembl Plants ? Try to explore the Ensembl Plants for more genes.
Response 2: Thank you very much for your suggestion. We chose NCBI as the main gene screening database because of the authority of the NCBI database, and many similar types of research articles have chosen the NCBI database (Wang et al., 2020; Fan et al., 2020; Su et al., 2021). We followed your suggestion and re-screened and identified the members of melon GLP gene family using the website of Ensembl plants (http://plants.ensembl.org/index.html). As you can see, using Ensembl plants to screen for the GLP results in exactly the same results we got. We have added this part of the work to section 2.2 and 3.1.
Fan, LIU.; Na, T.; Xueli, SUN.; et al. Genome-wide Identification and Expression Analysis of Banana GLP Gene Family. Acta Horticulturae Sinica. 2020. 47, 1930.
Su, W.; Zhang, C.; Feng, J.; et al. Genome-wide identification, characterization and expression analysis of the carotenoid cleavage oxygenase (CCO) gene family in Saccharum. Plant Physiol. Bioch. 2021, 162, 196-210.
Wang, J.; Zhang, Z.; Wu, J.; et al. Genome-wide identification, characterization, and expression analysis related to autotoxicity of the GST gene family in Cucumis melo L. Plant Physiol. Bioch. 2020. 155, 59-69.
Point 3: Extensive information are required in "section 2.2".
Response 3: Thank you for your suggestion. We have supplemented and modified the content of Section 2.2.
Point 4: Cite the all tools and databases in significant manner only URLs are not enough.
Response 4: Thanks for your suggestion. We have added the corresponding references.
Point 5: In section 2.5, define the how many genes you have derived from Arabidopsis and rice for phylogeny analysis? Authors are advised to collected more genes from evolutionary closed species to conduct extensive phylogeny analysis.
Response 5: Thank you very much for your suggestion. We added the Cucumber (Cucumber, like melon, belongs to the horticultural plant of Cucurbitaceae.) GLPs to the phylogenetic tree according to your opinion, and reconstructed the phylogenetic tree (Fig. 2, Line: 231). Also, we answered your question in section 2.5: finally, we used 43 rice GLPs, 32 Arabidopsis GLPs, and 38 cucumber GLPs to construct a phylogenetic tree (Lines: 136).
Point 6: Authors are advised to conduct the target miRNAs analysis
Response 6: Thank you very much for your suggestion. MicroRNAs (miRNAs) are a class of evolutionarily conserved endogenous non-coding single-stranded RNA molecules containing about 20-22 nucleotides and are important negative regulators of gene expression. We first downloaded the melon miRNA sequence from the miRNA database (http://structuralbiology.cau.edu.cn/PNRD/download.php). Then we used Bioinformatics & Systems Biology Online website (http://bioinformatics.psb.ugent.be/webtools/tapir/) to match melon miRNA sequence with melon GLP gene family members. Regrettably, we did not found any miRNAs that could target members of the melon GLP gene family. Thanks again for your suggestion, it will be a direction we need to pay attention to in future research.
Point 7: Evaluate the gene specific SSR markers potential of identified genes.
Response 7: Thank you very much for your suggestion. As an important branch in the field of biotechnology, DNA molecular marker technology is widely used in germplasm identification, assisted selection breeding, kinship identification and genetic diversity analysis. Among them, SSR molecular markers have some advantages: abundant quantity, high polymorphism, co-dominant inheritance, good repeatability, strong specificity and so on. We used the MISA website (https://webblast.ipk-gatersleben.de/misa/) to search and analyze the SSR loci of members of the melon GLP gene family. The results showed that only one SSR (TTC) sequence was predicted, which was located at the position of 18bp-34bp in the CDS of CmGLP12-1. We think this result is too singular to be included in the research results, I hope you can understand. Thanks again for your suggestion.
Point 8: Perform the protein structure modeling of representative proteins followed by structure evaluation using Ramachandran plot.
Response 8: Thank you very much for your suggestion. We followed your comments and added it in section 3.6 (Lines: 331-338).

Round 2
Reviewer 1 Report
Thank you for addressing my comments and concerns!
Reviewer 2 Report
Authors have significantly improved the quality and manuscript and addressed the all my comments.
Now, I strongly recommend this MS for publication in esteemed IJMS journal.